# The Search for the Pathogenic T Cells in the Joint of Rheumatoid Arthritis: Which T-Cell Subset Drives Autoimmune Inflammation?

**DOI:** 10.3390/ijms24086930

**Published:** 2023-04-08

**Authors:** Hisakata Yamada

**Affiliations:** Department of Clinical Immunology, Faculty of Medical Sciences, Kyushu University, 3-1-1 Maidashi, Higashi-ku, Fukuoka 812-8582, Japan; yamada.hisakata.579@m.kyushu-u.ac.jp

**Keywords:** rheumatoid arthritis, synovium, Th1, Th17, Tph, CTL

## Abstract

Rheumatoid arthritis (RA) is a chronic inflammatory disorder affecting systemic synovial tissues, leading to the destruction of multiple joints. Its etiology is still unknown, but T-cell-mediated autoimmunity has been thought to play critical roles, which is supported by experimental as well as clinical observations. Therefore, efforts have been made to elucidate the functions and antigen specificity of pathogenic autoreactive T cells, which could be a therapeutic target for disease treatment. Historically, T-helper (Th)1 and Th17 cells are hypothesized to be pathogenic T cells in RA joints; however, lines of evidence do not fully support this hypothesis, showing polyfunctionality of the T cells. Recent progress in single-cell analysis technology has led to the discovery of a novel helper T-cell subset, peripheral helper T cells, and attracted attention to the previously unappreciated T-cell subsets, such as cytotoxic CD4 and CD8 T cells, in RA joints. It also enables a comprehensive view of T-cell clonality and function. Furthermore, the antigen specificity of the expanded T-cell clones can be determined. Despite such progress, which T-cell subset drives inflammation is yet known.

## 1. Introduction: Implication of T Cells in the Pathogenesis of RA

Rheumatoid arthritis (RA) is a chronic inflammatory disorder affecting systemic synovial tissues, which results in progressive destruction of multiple joints. The etiology of RA is still unknown, but autoimmunity to certain synovial antigens is thought to be the underlying mechanism. Although clinical application of reagents targeting innate cytokines, such as tumor necrosis factor (TNF)-α and interleukin (IL)-6, has greatly improved the prognosis of the disabling disease, it does not cure the disease. In addition, increased susceptibility to infection is an inevitable side effect of blocking those cytokines. Therefore, development of treatment strategies regulating specific autoimmunity of RA is urgently needed.

Autoimmunity is an adaptive immune response toward self-antigens and is thereby mediated by T and B cells. Although the presence of disease-specific antibodies, such as rheumatoid factor (RF) and anti-citrullinated protein antibody (ACPA), implicates their involvement in disease onset, it has long been believed that autoreactive T cells, especially CD4 T cells, drive inflammation in RA joints. There are several findings that support this. Firstly, the typical histology of RA synovium, not only in the joint but also in the tendon sheath, shows massive infiltration of CD4 T cells [1]. Most of the CD4 T cells infiltrating the synovium express activation markers [2,3]. Secondly, there is a strong genetic association between certain major histocompatibility complex (MHC) class II alleles, the human leukocyte antigen (HLA)-DR “shared epitope”, and RA, especially seropositive RA [4]. Third, CD4 T cells are indispensable for the disease development of most animal models of RA [5]. Lastly, blocking T-cell costimulation with cytotoxic T-lymphocyte-associated antigen 4 (CTLA4)-Ig is clinically effective as a treatment for RA. The clinical efficacy of CTLA4-Ig is associated with ACPA positivity, and the presence of shared epitopes [6,7], suggesting that CD4 T cells but not CD8 T cells are the main target. 

Hence, efforts have long been made to identify the pathogenic autoreactive T cells in RA by analyzing clinical samples from the patients as well as studying experimental models of arthritis. Here, recent progress in single-cell analysis technology has greatly facilitated the understanding of the characteristics of various cell populations, including T cells, in the joints of RA patients. It sheds light on the previously unappreciated T cell subsets and provides comprehensive view on the clonality of T cells. There are also accumulating data about their antigen specificity. Integrating emerging information will be beneficial for future identification of the pathogenic T cells that drive the autoimmune inflammation in RA. Therefore, this review aims to summarize the up-to-date comprehensive knowledge about T cells in human RA-affected joints.

## 2. Traditional Hypothesis-Driven Research for Pathogenic T-Cell Subsets in RA: From Th1 to Th17

The prototype of T-cell-mediated inflammation is delayed-type hypersensitivity (DTH) reaction, also called type IV hyperreactivity, in which macrophages presenting the target antigen are activated by antigen-specific T-helper (Th) 1 cells that produce interferon (IFN)-γ. Because the histological feature of RA synovium resembles that of a DTH reaction, in that it shows an infiltration of activated CD4 T cells and macrophages, Th1 cells were widely accepted as the pathogenic T-cell subset in RA. The immunopathogenesis of RA, as well as its animal models, was often explained by the disturbed balance between Th1 and Th2. However, there are also conflicting notions. For instance, the expression of IFN-γ was low in RA joints [8,9], and mice lacking IFN-γ showed exacerbated arthritis [10].

Those findings even questioned the importance of T cells in the pathogenesis of RA [11], but T cells have again drawn attention due to the discovery of a novel helper T-cell subset in mice, Th17 cells [12], which produce IL-17 but not IFN-γ or IL-4. IL-17 is a proinflammatory cytokine involved in the mobilization of neutrophils and activation of fibroblasts, which results in the promotion of osteoclastogenesis via expression of the receptor activator of nuclear kappa-B ligand (RANKL) [13,14]. IL-17 is critical for the development of experimental models of arthritis [15,16] and was detected in the joints of RA patients [17,18]. The presence of human Th17 cells was first reported in patients with Crohn’s disease [19], while we and others were investigating Th17 cells in RA patients [20]. Unexpectedly, Th17 cells were not increased in the peripheral blood (PB) of RA patients compared to healthy controls, and there was no correlation between the frequency of Th17 cells and the disease activity of RA. Importantly, the frequency of Th17 cells was lower than that of Th1 cells in the joint, in contrast to the case of mouse models of RA [20,21]. The frequency of Th17 cells in the joint was even lower than that in PB. Consistent with these observations, it was reported afterward that IL-17-targeting therapy is not effective in human RA, in contrast to the cases of psoriasis and spondyloarthritis, other chronic inflammatory disorders of the joint [22]. Thus, Th17 cells do not play critical roles in the pathogenesis of RA, at least not by producing IL-17 in the established stage. It was also suggested that the different percentages of Th1 and Th17 cells in human RA and its mouse models might be due to species differences in the immune system or different pathogeneses of arthritis; in other words, the latter might be more like spondyloarthritis.

Another important issue regarding the difference between human and mouse Th17 cells is their ability to produce granulocyte–macrophage colony-stimulating factor (GM-CSF). GM-CSF is a proinflammatory cytokine involved in the pathogenesis of RA, as shown by clinical trials targeting GM-CSF [23]. GM-CSF is well known to be produced by innate immune cells, but CD4 T cells were reported to be an important cellular source of GM-CSF in RA joints [24]. In mice, GM-CSF is a critical effector cytokine defining the pathogenicity of Th17 cells [25]. However, in humans, a distinct helper T-cell subset producing GM-CSF was identified [26]. Furthermore, GM-SCF production in human T cells is linked to Th1 but is rather suppressed in Th17-inducing conditions in vitro [26]. We found that the majority of GM-CSF-producing cells in RA joints also produced IFN-γ, but not IL-17 [27]. Intriguingly, a substantial portion of the GM-SCF- and IFN-γ-double producing T cells also produced IL-21, indicating polyfunctionality of CD4 T cells in RA joints. We soon came to realize that IL-21 production is linked to a newly identified T-cell subset, as discussed in the following section. A recent study demonstrated an increased polyfunctionality of synovial CD4 T cells, as estimated by the expression of IL-2, TNF-α, IFN-γ, IL-17A, IL-22, IL-4, and GM-CSF, before the onset of RA [28]. Importantly, it was shown in mouse experiments that an in vivo immune response induces a continuum of helper T-cell phenotypes, rather than clearly segregated classical T-helper subsets [29], which might also apply to human T cells, including those in RA joints. Therefore, to understand the polyfunctional nature of T cells in RA joints, it is necessary to analyze T-cell functions in a more comprehensive manner than achieved by the classical studies that focus on a known cell population. This is now possible due to the new technologies emerging in this research field.

## 3. A Novel CD4 T-Cell Subset Identified by Emerging High-Dimensional Single-Cell Analysis: Tph Cells

Although the “classical” flow cytometric analysis provides much information on molecular expression at single-cell levels, it depends on a hypothesis-based selection of fluorochrome antibodies for detection. However, recent progress in comprehensive high-dimensional analyses, such as mass cytometry (cytometry by time of flight, CyTOF) and single-cell RNA sequence (scRNAseq), has enabled examination of the expression of huge numbers of molecules or genes in each cell without prior information, which has drastically changed many fields of biological research, including RA. By using CyTOF, Rao et al. identified a novel CD4 T-cell population expressing high levels of programed death-1 (PD-1), an inhibitory receptor induced in T cells continually stimulated with T-cell receptors (TCRs), in the joints of RA patients [30]. An increase in the PD-1 expression of T cells in RA joints was demonstrated even before the discovery of Tph cells [31,32], but these studies did not recognize PD-1 as a marker of distinct CD4 T-cell populations. PD-1^high^ CD4 T cells secrete IL-21 and C-X-C motif chemokine ligand (CXCL)13, a chemoattractant for B cells, and the ligand for C-X-C motif chemokine receptor (CXCR) 5, and are able to induce antibody production from B cells in vitro, similar to follicular helper T (Tfh) cells, which also express PD-1, as well as IL-21 and CXCL13, and help germinal center (GC) B cells in secondary lymphoid organs. However, the PD-1^high^ CD4 T cells in RA joint do not express CXCR5, unlike Tfh cells. Hence, the PD-1^high^ CXCR5^-^ CD4 T cells are called peripheral helper T (Tph) cells. A cluster of T cells corresponding to Tph cells was also detected by scRNAseq analysis of RA synovial cells [33]. A more recently developed analysis detects two populations of Tph cells, namely PD-1^high^ CXCL13^low^ cells and CXCL13^high^ cells in RA SF [34], but these are clonally related and, therefore, might reflect different activation statuses of the same subset of T cells.

There are also differences in the expression of transcription factors between Tfh and Tph cells. While Tfh cells express B-cell lymphoma 6 (BCL6) but not B-lymphocyte-induced maturation protein 1 (BLIMP1), the opposite trend is seen in Tph cells, although both express Maf (Table 1). In relation to this, Tfh and Tph cells are different in their ability to help naive B cells in vitro [35]. The development of human Tfh cells is experimentally induced by IL-12, Activin A, and transforming growth factor (TGF)-β [36]. As for Tph cells, Yoshitomi et al. have investigated their differentiation mechanism [37]. They first reported the presence of a T-cell population that spontaneously produces CXCL13 but is distinct from Tfh cell populations in RA joints, and named it Th_CXCL_13 [38], which is likely identical to the later discovered Tph. They subsequently found that in vitro culture of naive T cells in the presence of TCR stimulation and TGF-β induced CXCL13-producing CD4 T cells, and Sox4 was identified as the key transcription factor [37,39]. However, Sox4 failed to induce CXCL13 production in murine CD4 T cells in vitro [37], and there has been no study showing the presence of Tph cells in the joints of mouse models of RA. On the other hand, another group showed that in mice, CD4 T cells that had undergone lymphopenia-induced proliferation in vivo showed the PD-1^+^ CXCR5^-^ phenotype. These T cells helped B cells to produce antibodies via IL-21 secretion, similar to human Tph cells [40]. This might be an alternative mechanism of human Tph cell differentiation, although its relevance needs to be addressed.

As CXCL13 plays critical roles in GC formation, Tph cells are suggested to be involved in ectopic lymphoid neogenesis (ELN) in RA synovium. In line with this, Tph cells are abundant in the joints of ACPA-positive RA patients [30], and ACPA production by synovial B cells has been demonstrated [41]. However, it should be noted that ELNs are detected in the synovium of seronegative RA patients and those with even psoriatic arthritis [42]. Taken together with the notion that ACPA is detected before the onset of RA [43] and that the formation of ELNs correlates with local inflammation but not with autoantibody production [44], ELNs, as well as Tph cells, do not contribute critically to the production of ACPA in the serum of RA patients. 

Aside from their helper functions for antibody production, Tph cells are equipped with proinflammatory functions. Rao et al. reported the expression of IFN-γ mRNA in sorted Tph cells [30]. Others reported, before the discovery of Tph, the presence of CD4 T cells that produce IL-21 and TNF-α in RA SF [45]. As described above, we also noticed a portion of CD4 T cells in RA joints that co-produce IFN-γ, IL-21, and GM-CSF [27]. We extend this finding by showing that PD-1^high^ CD4 T cells, namely Tph cells, in RA joints can produce TNF-α, IFN-γ, and GM-CSF, in addition to IL-21 and CXCL13, although there is a heterogeneity in the profile of cytokine production [46]. Intriguingly, blocking PD-1 signaling in Tph cells enhanced cytokine production and induced self-MHC-dependent spontaneous proliferation in vitro [46]. Taken together, this suggests a role for Tph cells as proinflammatory effectors in RA synovitis, not only helping B cells in local antibody production but also aiding ELN formation. It should be noted that Tph cells are not specifically detected in RA joints but are also present in other inflammatory tissues, such as the salivary grand of patients with Sjogren syndrome [47]. An expansion of Tph cells was also observed in the PB of patients with systemic lupus erythematosus [48]. These findings suggest the involvement of Tph cells in a variety of immune pathogenies and the presence of a common mechanism for their development. Thus, the discovery of Tph cells, which was achieved by using the latest technology, provides novel insights into the role of T cells in the joints of RA patients. It might also change our understanding of immune systems and treatment strategies for various immune-mediated disorders.

## 4. The Emerging Importance of Cytotoxic CD4 and CD8 T Cells in RA Joints 

The high-dimensional single-cell analysis identified additional T-cell subsets in RA joints (Table 2). Fonseka et al. found an increase in the CD27^-^HLA-DR^+^ CD4 T cell population in the joints compared to the PB of RA patients [49]. Notably, these cells express higher levels of cytotoxicity-associated genes, including *PRF1*, *GZMB*, *GZMA*, and *GNLY* (genes for perforin-1, granzyme B, granzyme A, and granulysin, respectively) in addition to Th1-related genes, including *IFNG* and *TBX21* [49]. Another group showed an increase in eomesodermin-expressing CD4 T cells in RASF [50]. They later showed, in flow cytometric analysis, an increased expression of cytotoxic molecules, including GZMB, PRF1, Hobit, NKG7, and GPR56 proteins, on CD4 T cells in SF of ACPA^+^ RA patients [34]. A scRNAseq analysis of SF cells identified a cytotoxic CD4 T-cell cluster expressing higher levels of *NKG*, *GZMH*, *PRF1*, and *ZNF683* (Hobit) genes, while *GPR56* was highly expressed in Tph cells. An earlier study on the histology of RA synovium showed the presence of perforin-expressing CD4 T cells [51]. Thus, CD4 T cells equipped with cytotoxic functions can be pathogenic in RA joints, although the roles of cytotoxic activity in the inflammatory process as well as the types of target cells, which likely express MHC class II, remain to be determined.

CD8 T cells have largely been ignored in the field of RA research, mainly due to a lack of evidence supporting their involvement, in contrast to the case of CD4 T cells as described at the beginning of this review. However, CD8 T cells not only exist in the joints of RA patients; their frequency among T cells is actually higher in the joints than PB [52]. Most CD8 T cells in the joint show an activated phonotype [3,53,54]. Here, the results of high-dimensional analysis of RA synovial cells further attract attention to CD8 T cells. Three CD8 T cell subsets with different expression patterns of cytotoxic molecules, *GZMK*, *GZMB*, and *GNLY*, were identified by scRNAseq [33]. On the other hand, CyTOF analysis identified four subsets of CD8 T cells based on the expression patterns of HLA-DR, PD-1, and PD-1^-^HLA-DR^+^ populations, including *GZMK^+^GZMB^+^* effector T cells and *GNLY^+^GZMB^+^* cytotoxic T lymphocytes (CTLs) (Table 3) [33]. Notably, the expression levels of IFN-γ of all these CD8 T-cell subsets were higher than those of CD4 T cells in RA joints. More recent analysis additionally demonstrated clonal expansion of the GZMK^+^GZMB^+^ CD8 T-cell population [52]. GZMK^+^GZMB^+^ CD8 T cells have lower cytotoxic potential than GZMK^-^ GZMB^+^ CD8 T-cell populations. However, recombinant enzymatically active GZMK can activate synovial fibroblast to produce IL-6, C-C motif ligand (CCL)2 and reactive oxygen species in vitro [52]. Consistently, the culture supernatants of synovial CD8 T cells exerted similar effects on synovial fibroblasts, suggesting their potential as pathogenic T cells in RA. Interestingly, such CD8 T cells can be activated to produce IFN-γ in an antigen-independent manner, namely stimulation with IL-12 + IL-15 in vitro, and the profile of gene expression is similar to that of innate T cells, such as mucosal-associated invariant T (MAIT) and invariant natural killer T (iNKT) cells [52]. This remind us of our earlier observation in a mouse experiment that self-specific, innate-like CD8 T cells can be activated by IL-12 + IL-15 independently of TCR signaling [55], although the antigen specificity of CD8 T cells is unclear in the case of human RA. The high levels of IFN-γ production by CD8 T cells shed light on a role for IFN-γ as an important proinflammatory mediator of RA synovium, which was once thought not to be the case due to the emergence of Th17 cells. Notably, scRNAseq analysis of the synovial macrophages and fibroblasts of RA patients identified cell clusters with a signature of IFN stimulation in both cell lineages [33]. IFN-γ is also implicated in the generation of RANKL+ effector B cells in RA joints [56]. 

Regarding the PD-1^+^ CD8 T cells identified in the above study [33], we found them capable of producing IL-21 but CXCR5-negative, like Tph cells [57]. In line with this, an earlier study showed an association between the formation of ELN and the presence of CD40L-expressing CD8 T cells in the synovium [58]. In vivo depletion of human CD8 T cells in RA synovial explants that were implanted in severe combined immunodeficiency (SCID) mice resulted in disintegration of the GC structure in ELN [59]. Therefore, although less attention has been paid to them than CD4 T cells, CD8 T cells might also be a candidate for the pathogenic T cells in the joints of RA patients.

## 5. Clonality of T-Cell Subsets in RA Joints Revealed by Comprehensive TCR Analysis

Although RA has long been regarded as a T-cell-mediated autoimmune disease, it is still challenging to demonstrate autoreactivity of T cells in RA joints. Based on the assumption that pathogenic, autoreactive T cells likely expand in the joints of RA patients, the clonality of synovial T cells has been studied by examining TCRVβ usage [60,61,62] or analyzing the complementarity-determining region 3 (CDR3) sequence [63]. Yamamoto et al. developed a method to comprehensively evaluate TCR clonality, single-strand conformation polymorphism (SSCP) analysis, while others performed spectratyping of CDR3 length, both of which showed T-cell clonal expansion [64,65]. More recently, non-biased, next-generation sequencing was applied to comprehensively analyze T-cell clonality in RA joints. Klarenbeek et al. reported that the highly expanded clones were shared between different joints in the same patients [66]. Interestingly, there was a limited clonal overlap between synovial tissue and SF [67]. This might reflect different modes of T-cell infiltration and/or different extents of clonal expansion between the two compartments.

Because the expression of PD-1 in Tph cells might reflect chronic stimulation by self-antigens, we performed TCR deep sequencing of Tph cells from RA joints by comparing these with non-Tph cells from the same joint or PB T cells. There was little overlap in the expanded clonotypes between Tph and other subsets [46]. Interestingly, it was previously reported that the CDR3 length distribution was altered in T cells from ACPA^+^ synovium compared to ACPA^−^ RA or spondyloarthritis [68], which might reflect the different frequencies of Tph cells in the samples. More recently, scRNAseq analysis was utilized to evaluate the phenotypes of clonally expanded T cells in RA joints, since gene expression levels and TCR sequences of the same cells can be determined using this technique. Argyriou et al. analyzed the relationship between the extent of clonal expansion and the functional T-cell clusters in RA SF and found that the most of expanded clones were found within Tph, followed by regulatory, effector, and cytotoxic CD4 T cell clusters [34], which further implies the importance of the recently identified CD4 T cell subsets in RA joints (Table 2). Thus, the association between the functions and clonal expansion of T cells in RA joints has been revealed. However, even though their functions fit very with the pathogenesis of RA, it does not mean that the expanded clones are pathogenic autoreactive T cells, unless their autoreactivity is formally shown.

## 6. Recent Progress in Understanding Antigen Specificity of T Cells in RA Joints

The autoreactivity of T cells in RA joints can be addressed by examining their reactivity to candidate self-antigens. After the discovery of ACPA, most studies examined synovial T-cell reactivity to citrullinated proteins, including tenacin-C [69], alpha-enolase [70], and type II collagen [71]. In the latest one, Maggi et al. isolated a peptide/HLA-DR complex from the synovial tissue of RA patients and found 37 native and 6 citrullinated peptides, among which 6 peptides were able to stimulate CD4 T cells in the PB of RA patients in vitro [72]. Although there have been many studies on the antigen specificity of PB T cells in RA, T cells that induce inflammation locally in the joint and those that help B cells in APCA production in lymphoid organs do not necessarily recognize the same antigen. Nevertheless, citrullinated-peptide-specific CD4 T cells, even those in PB, mostly belong to the Th1 subset [72,73,74], suggesting their proinflammatory roles. At present, it is unknown whether Tph cells also recognize citrullinated protein antigens, although the autoreactivity of Tph cells in RA joints has been suggested, as described above. Accumulating evidence shows the presence of Tph cells in many inflammatory conditions [75], but the only target antigen identified so far is gluten in Celiac disease [76]. Identifying the antigens, which are either of self or foreign origin, as recognized by Tph cells in various conditions, including RA, will pave the way for understanding the roles of Tph cells in health and disease.

Recent studies reported antigen specificity of expanded T-cell clones in RA joints. Turcinov et al. performed scRNAseq analysis on synovial T cells in treatment-naive RA [77]. A diverse TCR repertoire of CD4 T cells, including Tph cells, was observed, but several expanded clones could also be detected. They re-expressed TCR from the expanded clones and established 81 cell lines. By testing the reactivity of the cell lines against panels of native and citrullinated self-peptides and viral peptides, they found that some TCR reacted to viral peptides, including human cytomegalovirus (CMV), Epstein–Barr virus (EBV), human herpes virus-2, and JC virus, but none responded to any citrullinated self-peptides. On the other hand, Zhang et al. analyzed a synovial TCRb sequence from 196 RA patients and controls on a public database and found the presence of clonal expansion and shared TCR motifs in RA patients with the lymphoid synovial pathotype [78]. Interestingly, many of the expanded clonotypes were similar to pathogen-antigen-specific sequences, including CMV, EBV, influenza virus, and also severe acute respiratory syndrome coronavirus 2 (SARS-CoV-2). They concluded this results from crossreactivity since the samples were collected before the emergence of SARS-CoV-2, although the TCRb sequence alone is not sufficient to determine antigen specificity. Notably, an expansion of T cells specific for pathogens, such as viruses, bacteria, and protozoa, in RA joints has already been shown [79,80,81]. Oligoclonal expansion of viral antigen-specific T cells was observed in different joints of the same patient [82]. Activated and/or memory pathogen-specific T cells might preferentially accumulate at the site of inflammation as they downregulate receptors for homing to lymphoid organs while upregulating adhesion molecules and receptors for chemokines expressed in response to inflammation. Thus, antigen-nonspecific factors are the determinants of cell migration per se. In addition, some antigens derived from pathogens that persist in the body might indeed be expressed in the joints. In any case, T-cell oligoclonality does not necessarily indicate their autoreactivity.

In addition, proliferation of T cells does not always indicate antigenic stimulation, because some T cells, if not all, can proliferate in an antigen-independent manner. The aforementioned GZMK^+^GZMB^+^ CD8 T cells in RA joint provide an example. It was also shown that RA synovial T cells resembled T cells that had been stimulated in vitro with a cocktail of cytokines comprising IL-2, IL-6, and TNF-α [83]. These cytokine-activated T cells (Tcks) induce contact-dependent TNF-α production from monocytes in a similar manner to true synovial T cells [83]. The precursor of Tcks are memory CD4 T cells expressing CD25 CD69 and HLA-DR [84], and Tcks exhibit chemotaxic activity toward RA synovial fibroblasts [85]. Thus, Tcks might be included in the T-cell subsets found in the joints of RA patients, and it is of interest to examine which one Tck corresponds to.

Developing a means by which T cells receiving antigenic stimulation can be detect in situ might lead to identification of autoreactive T cells in RA joints. In this regard, an observation made by Ashouri et al. might be a breakthrough [86]. Nur77 is encoded by an immediate early gene, *Nr4a1,* and is upregulated upon stimulation with TCR but not cytokines. They showed the expression of Nur77 firstly in arthritogenic T cells in mouse models and then in a portion of human CD4 T cells in RA synovium but not PB or osteoarthritis synovium. It would be interesting to know the functions and antigen specificity of these putative self-reacting T cells in RA joints. In summary, although researchers have found “potentially” autoreactive T cells via various approaches, further progress is still needed to identify the T cells reacting to self-antigens and driving synovitis in RA joints.

## 7. Conclusions and Future Perspective

This review summarized current knowledge on the function and antigen specificity of T cells in the joints of RA patients. Recent progress in single-cell analysis technology has allowed a comprehensive view of functional subsets and the clonality of synovial T cells. However, the fundamental question remains unanswered: which T cells induce and/or drive inflammation in RA joints? It is unlikely that all T cells in the joint are autoreactive and pathogenic. Rather, they include bystanders infiltrated as the results of inflammation. Therefore, identification of T cells actively responding to the autoantigen in situ, and not solely expressing activation markers or genes, is a critical step. Thereafter, antigen specificity may be addressed by cloning and re-expressing TCR. It is also important to further analyze the antigen specificity and phenotype of expanded T-cell clones, as already realized in several recent studies. Verifying the pathogenicity of autoreactive T cells is another challenging task. Although only clinical trials can provide formal proof, this could be addressed by in vitro experimental systems such as synovial tissue cultures. Further advances in basic research as well as in epidemiology will clarify the etiology of RA, leading to a cure for the disease.

## Figures and Tables

**Table 1 ijms-24-06930-t001:** Similarities and differences between Tfh and Tph cells.

	Tfh	Tph
**Similarity**		
Surface molecules	PD-1^high^, ICOS^+^	
Cytokines	IL-21, CXCL13	
Transcription factors	Maf	
**Difference**		
Location	Lymphoid organs	Inflamed tissues
Chemokine receptors	CXCR5^+^, CCR2^−^	CXCR5^−^, CCR2^+^
Transcription factors	BCL6	BLIMP1, Sox4

PD-1: programmed death-1, ICOS: inducible costimulator, IL-21: interleukin-21, CXCL13: C-X-C motif chemokine ligand 13, CXCR5: C-X-C motif chemokine receptor 5, CCR2: C-C motif ligand 2, BCL6: B-cell lymphoma 6, BLIMP1: B-lymphocyte-induced maturation protein 1.

**Table 2 ijms-24-06930-t002:** CD4 T-cell clusters in RA joints reported by Argyriou et al. [34].

Cluster Name	Differentially Expressed Genes	Frequency in SF
Naive CD4	TCF7, CCR7, LEF1	− ^1^
CXCL13high Tph	TNFRSF18, LAG3, CXCL13	+++
Central memory CD4	LTB, ZFP36L2, KLF2	±
Effector CD4	CXCR3, TGFB1, KLRB1	++
Treg	FOXP3, IL2RA, TIGIT	++
Cytotoxic CD4	NKG7, GNLY, GZMH	+
SESN3 CD4	TNFAIP3, SLC2A3, CDC14A	±
CXCL13low Tph	PTPN13, PRDM1, NEAT1	+
Humanin CD4	MT-ATP6, MT-ND4, MTRNR2L12	− ^1^
EGR1 CD4	EGR1, IER2, NR4A1	− ^1^
Proliferating CD4	STMN1, MKI67, TUBA1B	+
Activated CD4	CST3, HLA-DRA, HLA-DPA1	±

^1^ Note that several clusters were detected only in PB.

**Table 3 ijms-24-06930-t003:** CD8 T-cell clusters in RA joints reported by Zhang et al. [33] and Jonsson et al. [52].

scRNAseq Clusters	CyTOF Clusters	DEGs	Abundance
*GZMK^+^GZMB^-^*	PD-1^+^HLA-DR^+/*−*^	*GZMK*, *NKG7*	Joint = PB
*GZMK^+^GZMB^low^* (effector)	PD-1*^−^*HLA-DR^++^	*IFNG*, *HLA-DRB1*	Joint > PB
*GZMK^−^GZMB^high^* (cytotoxic)	PD-1*^−^*HLA-DR^+^	*PRF1*, *GNLY*	Joint < PB
*GZMK^−^GZMB^−^* (naive)	PD-1*^−^*HLA-DR*^−^* (?) ^1^	*CCR7*, *IL7R*	Joint < PB

^1^ No information provided.

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
