# Peer review of "The Search for the Pathogenic T Cells in the Joint of Rheumatoid Arthritis: Which T-Cell Subset Drives Autoimmune Inflammation?"

_ijms, 2023, doi:10.3390/ijms24086930_

Round 1
Reviewer 1 Report
In the current review, the author summarizes the evidence for the role of pathogenic T cells in the joint of rheumatoid arthritis and addresses a specific question on which T cell subset is the culprit for triggering the autoimmune inflammation response. The author lists prior studies on the implication of T cells in the pathogenesis of RA, traditional hypothesis-driven research for pathogenic T cell subsets in RA, a novel CD4 T cell subset identified by emerging high dimensional single cell analysis: Tph cells, presence of cytotoxic CD4 and CD8 T cells in the joint of RA, detection of clonally expanded T cells in RA joint, and antigen-specificity of T cells in RA joint. The authors provided two interesting figures that summarize the different abundance of Th1 and Th17 cells in human RA and its mouse models and similarities/differences between Tfh and Tph cells. From this discussion, the authors suggest that recent progress in single-cell analysis technology contributed to understanding comprehensive view on functional subsets and clonal expansion of synovial T cells. Yet, the fundamental question remains unanswered as to which T cells induce and/or drive inflammation in RA joint. While the summarized information is interesting and the review is well-written, the author does not provide take-home messages at the end of the respective section for the reader.
Comments:
1) The current title reads as “The search for the pathogenic T cells in the joint of rheumatoid arthritis: who drives the autoimmune inflammation?”. It does not reflect the main point of the present work and how it is unique relative to previous reviews on the same topic. Thus, to attract the interest of more readers and to make it clearer for readers, the authors may consider modifying the title to “The search for the pathogenic T cells in the joint of rheumatoid arthritis: Which T cell subset drives the autoimmune inflammation?”.
2) The provided sections read like narration for the evidence of discussed points without critical aspects/reflection points. At the end of each section, a take-home message is advised to be provided.
3) To avoid readers’ confusion, the authors are advised to clearly describe in the narration of previous literature whether these data are derived from clinical studies or from experimental studies. This point needs to be carefully addressed by the authors in the entire manuscript.
4) In section 1 (Introduction: Implication of T cells in the pathogenesis of RA), the authors are advised clearly specify the aim behind the current review and to clarify this point for readers by elaborating on how the current study is unique relative to the previous literature that addressed the same topic.
5) In the main text, please explain each abbreviation only once, when used for the first time, then use an abbreviation consequently. For example, in line 113, the PD-1 abbreviation was mentioned without explaining its full name. This point needs to be carefully addressed by the authors for all the used abbreviations in the entire manuscript.
6) The authors may need to add a list of abbreviations.
7) The authors are advised to make the figure captions stand-alone. To this end, authors are advised to provide the full names of all the listed abbreviations in the figures. For example, in figure 1, the full names of PD-1, ICOS+, IL-20, XCXL13, etc. need to be described.
8) More attractive colored version of figures is advised to enhance the readership of the present review.
9) A few typos/syntax are present in the manuscript which need to be addressed, for example:
A) In line 262: the authors state “Although this rewiew focuses …”. Please, correct it to “basic” to “Although this review focuses …”.
B) In lines 14-15: the authors state “Historically, Th1 and Th17 cells were hypothesized as the pathogenic T cells in RA joints, but lines of evidence do not fully support the hypothesis but rather show polyfunctionality of the T cells.”
Please, consider modifying it to “Historically, Th1 and Th17 cells were hypothesized as the pathogenic T cells in RA joints, however, lines of evidence do not fully support the hypothesis but rather show polyfunctionality of the T cells.”
C) In lines 10-11: the authors state “which is supported by a handful of basic as well as clinical observations”
Please, modify “basic” to “experimental”.
Author Response
1) The current title reads as “The search for the pathogenic T cells in the joint of rheumatoid arthritis: who drives the autoimmune inflammation?”. It does not reflect the main point of the present work and how it is unique relative to previous reviews on the same topic. Thus, to attract the interest of more readers and to make it clearer for readers, the authors may consider modifying the title to “The search for the pathogenic T cells in the joint of rheumatoid arthritis: Which T cell subset drives the autoimmune inflammation?”.
Reply:
Line 3, I changed the title of the manuscript following the reviewer's advice.
2) The provided sections read like narration for the evidence of discussed points without critical aspects/reflection points. At the end of each section, a take-home message is advised to be provided.
Reply:
I added following sentences as take-home messages at the end of sections:
Line 118-120: Therefore, to understand the polyfunctional nature of T cells in RA joints, it is necessary to analyze T cell functions in more comprehensive manners than the classical studies that focus on a known cell population.
Line 193-196: Thus, the discovery of Tph cells, which was achieved by using the latest technology, provides novel insights on the role of T cells in the pathogenesis of RA. It might also change our understanding on immune systems and treatment strategy for immune-mediated disorders.
Line 210-213: Thus, CD4 T cell equipped with cytotoxic functions can be pathogenic in RA joint, although the roles of cytotoxic activity in the inflammatory process as well as the types of target cells, which likely express MHC class II, remain to be determined.
Line 253-254: Therefore, although less attention has been paid than CD4 T cells, CD8 T cells might also be a candidate of the pathogenic T cells in the joint of RA patients.
Line 282-285: Thus, the association between the functions and clonal expansion of T cells in RA joints has been revealed. However, even though their functions fit very with the pathogenesis of RA, it does not mean that the expanded clones are the pathogenic autoreactive T cells, unless their autoreactivity is formally shown.
Line 344-347: In summary, although researchers have found "potentially" autoreactive T cells via various approaches, it is still on the way to identify the T cells reacting to self-antigens and driving synovitis in RA joints.
3) To avoid readers’ confusion, the authors are advised to clearly describe in the narration of previous literature whether these data are derived from clinical studies or from experimental studies. This point needs to be carefully addressed by the authors in the entire manuscript.
Reply:
Thank you for your comment on the point that we had not noticed. To avoid the confusion, we added words, such as "experimental", "clinically", and "in vitro" where applicable as follows:
Line 11, experimental; line 46, clinically; line 78, human; line 86, experimental; line 89, human; line 106, in vitro; line 152, experimentally; line 160, in vitro; line 233, in vitro; line 239, experiment; line 292, in vitro; line 329, in vitro.
4) In section 1 (Introduction: Implication of T cells in the pathogenesis of RA), the authors are advised clearly specify the aim behind the current review and to clarify this point for readers by elaborating on how the current study is unique relative to the previous literature that addressed the same topic.
Reply:
Following the reviewer's advice, the last part of the introduction was modified
Line 49-58: Hence, efforts have long been made to identify the pathogenic autoreactive T cells in RA by analyzing clinical samples from the patients as well as studying experimental models of arthritis. Here, recent progress in single cell analysis technology greatly facilitates understanding the characteristics of various cell populations, including T cells, in the joint of RA patients. It has unveiled previously unappreciated T cell subsets and provided comprehensive view on their clonality. There are also accumulating data about their antigen specificity. Integrating those emerging information will be beneficial for future identification of the pathogenic T cells that drive the autoimmune inflammation in RA. Therefore, this review aims to summarize the up-to-date comprehensive knowledge about T cells in human RA joints.
5) In the main text, please explain each abbreviation only once, when used for the first time, then use an abbreviation consequently. For example, in line 113, the PD-1 abbreviation was mentioned without explaining its full name. This point needs to be carefully addressed by the authors for all the used abbreviations in the entire manuscript.
6) The authors may need to add a list of abbreviations.
I spelled out the abbreviations once used for the first time and provided the list before the references (line 366-379).
7) The authors are advised to make the figure captions stand-alone. To this end, authors are advised to provide the full names of all the listed abbreviations in the figures. For example, in figure 1, the full names of PD-1, ICOS+, IL-20, XCXL13, etc. need to be described.
Reply:
I gave the list of abbreviations in the figure.
8) More attractive colored version of figures is advised to enhance the readership of the present review.
Reply:
I made colored version of the figures.
9) A few typos/syntax are present in the manuscript which need to be addressed, for example:
- A) In line 262: the authors state “Although this rewiew focuses …”. Please, correct it to “basic” to “Although this review focuses …”.
Reply:
I modified the whole sentence (line 292-293).
- B) In lines 14-15: the authors state “Historically, Th1 and Th17 cells were hypothesized as the pathogenic T cells in RA joints, but lines of evidence do not fully support the hypothesis but rather show polyfunctionality of the T cells.”
Please, consider modifying it to “Historically, Th1 and Th17 cells were hypothesized as the pathogenic T cells in RA joints, however, lines of evidence do not fully support the hypothesis but rather show polyfunctionality of the T cells.”
Reply:
I changed the text as the reviewer suggested (line 14-16).
- C) In lines 10-11: the authors state “which is supported by a handful of basic as well as clinical observations”
Please, modify “basic” to “experimental”.
Reply
I modified the sentence as well as wording (line 11-12).
Reviewer 2 Report
I have had the pleasure of reviewing "The search for ... autoimmune inflammation?" by Yamada H. The review outlines the contributions of T cell subsets to RA pathology and includes Th1, Th17, Tfh, Tph etc; there is also discussion of the detection of T cell subsets and the nature of auto-reactivity.
Scope
The scope of the review is very wide and would benefit from refinement. Each of the cell subsets mentioned might provide sufficient material for review.
Language
Although the author's intent is clear, some of the phrasing is atypical and some errors are present.
Ten are listed below as examples.
E.g.
p1 Ln3 "who drives the inflammation" - Was the use of 'who' deliberate?
p3 ln 110
p3 Ln 115 cell/has should be cells/have
p3 Ln 121 'neither' is not the correct word
p3 Ln 129 'the ability of helping naive but not memory B cells is different' this is awkwardly phrased
p3 Ln 137 enhances
p6 Ln 262 rewiew
p7 Ln 315 'knowledges on the functions'
p7 Ln 317 'contributes to understanding comprehensive view'
p7 Ln 320 'They rather' might be rephrased as 'Rather, they...'
Figures
Figure 1 does not add much to the submission.
Conclusion
From Ln 321 onwards the author identifies some key points regarding improving the understanding of T cells in autoimmune disease. It would improve the submission if this could be expanded upon, with examples of how this might be acheived.
Prior work
The submission bears some similarity to [Hisakata Y. (2022) Immunological Medicine, 45:1, 1-11]. Some editing might be necessary to further differentiate this submission from the previously published work.
Author Response
I have had the pleasure of reviewing "The search for ... autoimmune inflammation?" by Yamada H. The review outlines the contributions of T cell subsets to RA pathology and includes Th1, Th17, Tfh, Tph etc; there is also discussion of the detection of T cell subsets and the nature of auto-reactivity.
Scope
The scope of the review is very wide and would benefit from refinement. Each of the cell subsets mentioned might provide sufficient material for review.
Reply:
Thank you very much for your comments. I agree that this review describes on a wide range of subjects, which might give somewhat scattered impression. However, recent introduction of high-dimensional single cell technology has rapidly and greatly changed the style of research. As for the field of arthritis research, it provides unbiased comprehensive view of the entire cell populations, including T cells, in human joints. I think it is urgently required to share knowledge about this with the readers, and this review might be a good opportunity. In addition, all T cell subsets that have been identified by those studies are attractive candidates for the pathogenic T cells in RA. It is another reason why I did not just focus on a T cell subset. Lastly, most of the studies that are introduced in this review compared T cell populations in RA joints with those in peripheral blood or in normal joints. Thus, it can be said that these studies try to determine the etiology of RA by analyzing the changes of cell populations, and I understand that "change" is the keyword of this Special Issue. Therefore, I revised the manuscript so that the readers can understand the aim of this review by adding explanation in the introduction as follows (Line 49-58):
Hence, efforts have long been made to identify the pathogenic autoreactive T cells in RA by analyzing clinical samples from the patients as well as studying experimental models of arthritis. Here, recent progress in single cell analysis technology greatly facilitates understanding the characteristics of various cell populations, including T cells, in the joint of RA patients. It has unveiled previously unappreciated T cell subsets and provided comprehensive view on their clonality. There are also accumulating data about their antigen specificity. Integrating those emerging information will be beneficial for future identification of the pathogenic T cells that drive the autoimmune inflammation in RA. Therefore, this review aims to summarize the up-to-date comprehensive knowledge about T cells in human RA joints.
Language
Although the author's intent is clear, some of the phrasing is atypical and some errors are present.
Ten are listed below as examples.
E.g.
p1 Ln3 "who drives the inflammation" - Was the use of 'who' deliberate?
Reply:
I changed to "Which T cell subset drives the autoimmune inflammation?"(line 3)
p3 ln 110
p3 Ln 115 cell/has should be cells/have
Reply:
I changed "cell" to "cells"; I am afraid that "has" is applicable in this case (line 134)
p3 Ln 121 'neither' is not the correct word
Reply:
I changed the sentence to use "do not express" (line 141)
p3 Ln 129 'the ability of helping naive but not memory B cells is different' this is awkwardly phrased
Reply:
Tfh and Tph cells are different in the ability to help naive, but not memory, B cells in vitro (line 150)
p3 Ln 137 enhances
Reply:
I would like to change it to "enhanced"(line 158)
p6 Ln 262 rewiew
Reply:
I do not use the word "review" there anymore (line 292).
p7 Ln 315 'knowledges on the functions'
Reply:
I changed to "knowledge on the function" (line 349)
p7 Ln 317 'contributes to understanding comprehensive view'
Reply:
I changed "contributes to understanding" to "provides" (line 351)
p7 Ln 320 'They rather' might be rephrased as 'Rather, they...'
Reply:
I changed as suggested (line 354).
Figures
Figure 1 does not add much to the submission.
Reply:
We modified the figure to be more informative. We added explanation in the manuscript text accordingly (line 92-95):
It was also suggested that the different predominance of Th1 and Th17 cells in human RA and its mouse models might be due to species difference in the immune system or different pathogenesis of arthritis, i.e., the latter might be more like spondyloarthritis (FIgure 1).
Conclusion
From Ln 321 onwards the author identifies some key points regarding improving the understanding of T cells in autoimmune disease. It would improve the submission if this could be expanded upon, with examples of how this might be acheived.
Reply:
Following the reviewer's advice, I added more detailed description on this issue as follows (Line 356- 359):
Thereafter, antigen-specificity may be addressed by cloning and re-expression of TCR. It is also important to further analyze antigen-specificity and phenotype of expanded T cell clones, as already done in some recent studies.
Prior work
The submission bears some similarity to [Hisakata Y. (2022) Immunological Medicine, 45:1, 1-11]. Some editing might be necessary to further differentiate this submission from the previously published work.
Reply:
I edited the sentences throughout the manuscript to decrease similarity with those in the prior publication as much as possible. Although there are some overlaps in the subjects, especially about earlier studies, we revised the manuscript to emphasize and focus more on recent ones, including the subheadings.
Lines 38-40, 4-45, 73-74, 80-81, 83-84, 88-90, 106-108, 118-121, 193-197, 255-260, 263-264, 266-270, 282-286, 292-293, 296-297, 300-302, 332-335, 337-339.
Round 2
Reviewer 2 Report
Previously published work
As mentioned in the previous review, the current submission appears to subtantially overlap with a previous publication; see below.
Figures
The figures have not significantly changed.
Figure 3 might better be conveyed as a table.
Author Response
Dear Reviewer,
Thank you again for your comments on my manuscript. I have modified the manuscript following your advice as follows:
Previously published work
As mentioned in the previous review, the current submission appears to subtantially overlap with a previous publication;
Reply:
I have extensively edited throughout manuscript to reduce overlaps.
Figures
The figures have not significantly changed. Figure 3 might better be conveyed as a table.
Reply:
I have deleted Figure 1 as well as Figure 2. Figure 2 has been changed to a table (Table 1), and I have also added a new table about CD8 T cell subsets as table 3.
Round 3
Reviewer 2 Report
The author has significantly changed the text of the submission such that it does not overlap with the previously published work of the author.
Figures/Tables have been improved.